# Establishing a Herpesvirus Quiescent Infection in Differentiated Human Dorsal Root Ganglion Neuronal Cell Line Mediated by Micro-RNA Overexpression

**DOI:** 10.3390/pathogens11070803

**Published:** 2022-07-16

**Authors:** Yu-Chih Chen, Hedong Li, Miguel Martin-Caraballo, Shaochung Victor Hsia

**Affiliations:** 1Department of Pharmaceutical Sciences, School of Pharmacy and Health Professions, University of Maryland Eastern Shore, Princess Anne, MD 21853, USA; ychen@umes.edu (Y.-C.C.); mmartin@umes.edu (M.M.-C.); 2Department of Neuroscience & Regenerative Medicine, Medical College of Georgia, Augusta University, 1120 15th Street, Rm. CA4012, Augusta, GA 30912, USA; hedli@augusta.edu

**Keywords:** host–virus interaction, herpesvirus, microRNA, dorsal root ganglion, latency, reactivation

## Abstract

HSV-1 is a neurotropic pathogen associated with severe encephalitis, excruciating orofacial sensation, and other chronic neuropathic complications. After the acute infection, the virus may establish a lifelong latency in the neurons of trigeminal ganglia (TG) and other sensory and autonomic ganglia, including the dorsal root ganglia (DRG), etc. The reactivation occurred periodically by a variety of physical or emotional stressors. We have been developing a human DRG neuronal cell-culture model HD10.6, which mimics the mature neurons for latency and reactivation with robust neuronal physiology. We found that miR124 overexpression without acyclovir (ACV) could maintain the virus in a quiescent infection, with the accumulation of latency-associate transcript (LAT). The immediate-early (IE) gene ICP0, on the other hand, was very low and the latent viruses could be reactivated by trichostatin A (TSA) treatment. Together, these observations suggested a putative role of microRNA in promoting HSV-1 latency in human neurons.

## 1. Introduction

Herpes Simplex Virus Type 1 (HSV-1) is a shrewd pathogen, which infects an extensive range of human cell types, resulting in high human morbidity. Herpes Simplex Encephalitis (HSE) is sporadic but quite lethal with a mortality rate of 70–90% if left untreated [1]. Furthermore, over half of its survivors experience neurological deficits [2]. HSV-1 is responsible for a variety of ocular complications due to reactivation from latently infected neurons in the trigeminal ganglia (TG) [3,4]. Epidemiological data indicated that approximately 50,000 new and recurrent cases of HSV infections are reported annually in the United States [5].

Human microRNA-124 (hsa-miR-124 or miR-124) is one of the most abundant microRNAs in neurons [6]. It plays critical roles in neurogenesis [7,8], synaptic functions [9,10], and cognitive impairment [11], as well as cancers [12]. The miR-124 precursor is a non-coding RNA molecule that has been identified in a variety of species, including humans. Its mature form, approximately 22 nucleotides, is processed from the hairpin precursor by the Dicer enzyme. There is no report linking miR-124 to HSV-1 biology, although it was used to inhibit non-specific replications of oncolytic HSV gene therapy vectors to treat glioblastoma [13]. On the other hand, it was reported that human cytomegalovirus (HCMV) latent infection alters the expression of cellular and viral microRNAs [14]. It appeared that the expression of human miR-124-3p significantly increased in the HCMV latent infection library. In addition, hsa-miR-124, which modulated macrophage activation, increased during HCMV latent infection. By blocking the macrophage’s activation, hsa-miR-124 assisted in creating a cellular environment for HCMV latent infection [15].

HSV latency studies have been performed primarily in rodent models. Due to physiological differences, developing a human model to comprehend the viral pathogenesis of how HSV establishes latency and how it induces reactivation in human neurons is necessary [16,17]. Human neurons in culture die quickly after HSV-1 infection unless the replication inhibitor acyclovir (ACV) is present, probably because cultured neurons are missing some important expression of neuronal-specific genes due to the lack of miRNA. We hypothesize that, based on a previous publication [6], miR-124 is involved in reinforcing neuronal differentiation and, thus, supports HSV-1 quiescent infection in neurons.

## 2. Results

### 2.1. The miR124 Expression Increased in HD10.6 Cells

The expression of mature miR-124 by L124 transduction (miR-124-loaded lentiviral vector, described in Materials and Methods Section 4.3) was first validated in HEK-293 cells. The results of qRT-PCR showed an approximately 10-fold increase from L124 in comparison to vector control (Figure 1A). Then, we transduced miR-124 into HD10.6 cells with L124 and it demonstrated a 4.95-fold and 3-fold increase in the absence of or presence of HSV-1 infection, respectively (Figure 1B).

### 2.2. HSV-1 Can Infect Differentiated HD10 Cells with miR124 Overexpression

The HD-L-Stable cells infected with KVP-mRFP HSV (described in Materials and Methods Section 4.4) were observed under a fluorescent microscope at 3 dpi. The robust green fluorescence indicated that miR124 was successfully introduced and expressed in HD-L-Stable cells (Figure 2). The detection of red fluorescence revealed a successful HSV-1 infection (Figure 2). The merged image indicated that cells had a miR-124 expression with HSV-1 infection simultaneously.

### 2.3. The miR124 Expression Reduced the Size of the HD10.6 Cells

HSV infection causes a reduction in the electrical excitability of infected neurons [18]. To assess how HSV infection in the presence of miR124 alters the electrical excitability of HD10.6 cells, we performed whole-cell recordings of voltage-activated sodium currents. Sodium currents were normalized to the cell size as determined by measurements of cell capacitance [18]. Note that the plasma membrane behaves similarly to a capacitor, and the phospholipid bilayer acts as a thin insulator, dividing two electrolytic media, in the extracellular and intracellular environments. The capacitance is, thus, directly proportional to the cell surface area. The measurement of cell capacitance revealed that HD-L-Stable cells exhibited approximately a 40% decrease in terms of their sizes (Figure 3A). Interestingly, there were no significant changes in the sodium channel current density of HD-L-Stable cells compared to the controls (Figure 3B). However, an HSV-1 infection of HD-L-Stable cells caused a significant reduction in the density of sodium currents (Figure 3B). These findings suggest that HSV-1 infection is still capable of reducing the expression of sodium channels even in HD-L-Stable cells.

### 2.4. The Overexpression of miR124 Promoted a Quiescent State of HSV-1 Infection

Differentiated normal HD10.6 and the HD-L-Stable cells were initially infected by HSV-1 at the MOI of 1 in the absence of ACV. The daily release of infectious viruses was measured by plaque assays using the generated media supernatant. It appeared that the HSV-1 replicated actively in the normal cells and viral particles were released vigorously (Figure 4A), and these cells died at 4 dpi (data not shown). While in HD-L-Stable cells, there were fewer viral particles released compared to the regular counterpart (Figure 4A) and the cells remained healthy until 7 dpi (data not shown). To create a dormant state of infection, the MOI was reduced to 0.15 and the release of infectious viral particles was not detected until 6 dpi in the normal HD10.6 cells (Figure 4B). As for the HD-L-Stable cells, there were no detections of any infectious virus release (Figure 4B). It is worth mentioning that both cells remained healthy throughout the experiments.

### 2.5. HSV-1 Replication Occurred but Failed to Be Released from the HD-L-Stable Cells

To induce viral reactivation, the quiescent infection was treated with histone deacetylase inhibitor TSA [18]. There was no release of infectious viruses in the HD-L-Stable cells (Figure 5A). To further investigate the virus progeny trapped within the cells, the cell lysates were prepared for plaque assays, and it showed that HD-L-Stable cells contained some viruses with a 3.46-fold reduction compared to the normal cells (Figure 5B). The viruses trapped inside the HD-L-Stable cells retained the capability to infect Vero cells and the TSA increased replication by 2.47-fold (Figure 5B). These results demonstrated that viral replication was present but it was reduced and restricted in HD-L-Stable cells.

### 2.6. TSA Increased the Viral Gene Transcription from the Quiescent State of Infection

We continued to use TSA to reactivate latently HSV-1 infected HD10.6 cells, and viral gene expressions were assessed by qRT-PCR [19]. Before TSA treatment, the transcripts of the viral genes TK and ICP0 were very low but detectable in the HD-L-Stable cells (Figure 6A,B). TSA treatment induced a 7.2-fold and 7.8-fold increase in TK and ICP0 expression, respectively (Figure 5A,B). The expression of miR124, however, remained unchanged following TSA treatment (Figure 6C).

### 2.7. HSV-1 LAT Accumulated in the miR124 Expressing HD10.6 Cells

The HSV-1 LAT accumulation was assessed by infecting the cells at an MOI of 0.15 followed by comparing between regular and HD-L-Stable cells. There was no difference in LAT expression levels at 3 dpi; but at 6 dpi, a significant LAT increase was displayed in HD-L-Stable cells (Figure 7A). The level of LAT in regular HD10.6 cells declined quickly, which is similar to what we observed previously [19]. The LAT expression after TSA treatment was analyzed and compared by qRT-PCR. It appeared that TSA, although it strongly increased TK and ICP0 expression, did not produce any significant change in the level of LAT (Figure 7B). In addition, HD-L-Stable cells displayed an estimated 4-fold increase in the expression of LAT compared to regular cells at 7 dpi (Figure 7B). These results suggested a role of miR124 in the maintenance of LAT during latency.

## 3. Discussion

In fully permissive non-neuronal cultured cells, HSV-1 encodes around 90 unique transcriptional units and completes the entire life cycle in less than 20 h [20] However, in latently infected neurons, the robust viral gene expression is silent and the only detectable viral gene products are a group of non-coding RNA. The molecular mechanism involved in these processes is still being investigated [19]. There is no canonical definition of latency. The “lexicon” of latency is that a viral genome stays and endures within a cell population for a time during which it maintains a dormant state without detectable levels of lytic viral transcripts, proteins, or infectious particles, yet it retains the capacity to replicate and reactivate under certain external clues [21,22,23,24,25,26,27,28]. So far, the animal models recapitulating these features of HSV latency would be those of mice and rabbits [29,30,31]. The rodent cell culture model, although it did not satisfy all characteristics of latency, offered benefits in studying the molecular mechanisms of HSV latency [18,26,31,32,33,34,35,36]. However, the lack of a human model poses a critical barrier for better understanding HSV-1 latency.

Our current human model was originally developed to study nociceptive properties and later repurposed to investigate HSV-1 gene regulation during latency [18,21,37]. It has, therefore, rendered this system as a novel model for studying latency and reactivation of all α-herpesviruses, particularly under the influence of neuronal excitation and epigenetics. However, HSV-1 kills human neurons in cell culture quickly in the absence of ACV. Furthermore, it is puzzling that LAT accumulates in the ACV-treated, quiescent-infected HD10.6 cells for as many as 3 days, followed by a decline to very low levels [18]. This may occur, most likely, because cultured neurons, during cell differentiation, are missing some important expression of neuronal-specific genes to suppress viral replication and promote LAT accumulation [6,38]. We hypothesize that this missing factor is miR-124. Therefore, the introduction of miR-124 into the proliferating neurons followed by differentiation would restore the missing expression of critical neuronal-specific genes, suppress viral replication, and support a dormant infection mimicking latency in the absence of ACV. In this case, the quiescent state of infection was achieved and can be maintained for as many as two weeks with a continuous presence of LAT throughout the process.

MicroRNAs primarily suppressed translation by binding to the 3′ untranslated regions of the mRNA targets [39,40]. Out of all of these, miR-124 is not only conserved from Caenorhabditis elegans to humans but also abundant in adult as well as embryonic neurons in the CNS [7,41]. It has been shown that the expression of miR-124 in HeLa cells decreases 174 genes expressed at a lower level in the brain [42]. This observation suggested the idea that miR-124 may regulate neuronal identity by blocking the transcription of non-neuronal genes in neurons. The exact mechanisms are not clear but the statement of “blocking non-neuronal genes in neurons” reminded us of another important transcription regulator REST/NRSF, which suppresses neuronal genes in non-neuronal cells [43].

REST/NRSF exerted its function by binding to a conserved neuronal gene loci repressor element (RE1). It assembled a corepressor complex by recruiting histone deacetylases and methyl CpG-binding protein MeCP2 [44,45]. A successful transition from progenitors to post-mitotic mature neurons was achieved by a down-regulation of REST/NRSF, which permitted neuronal gene expression [46]. To accomplish the process, REST/NRSF is supposed to be abundant in non-neuronal cells but absent in neurons. However, previous studies indicate that REST/NRSF, although absent in neurons of the brain, was observed in most neurons of murine trigeminal ganglia [47]. In addition, REST/NRSF inhibited miR-124 expression in non-neuronal and neural progenitor cells [48]. It is of interest to note that an anti-neural factor small C-terminal domain phosphatase 1 (SCP1), expressed in non-neuronal tissues, was recruited to RE1-containing neural genes by REST/NRSF [49]. The microRNA miR-124, during embryonic CNS development, was sufficient to antagonize the REST/SCP1 pathway [50]. Nonetheless, the conundrum is that the REST/NRSF/CoREST complex per se may participate in latency establishment, displaying activities that suppress HSV-1 gene expression [51,52].

It is unclear whether miR124 suppresses HSV-1 gene expressions directly. We performed a BLAST analysis [53] searching for putative binding sites of miR124 functional domains [54] on the HSV-1 genome and obtained several good hits (unpublished results). For example, two hits matched with the HSV-1 UL 36 gene in which there were both perfect matches: cDNA sequence 1545–1556 and 4232–4241 (Figure 8). UL36 is a gamma 2 protein, and it is essential for virion egress through the cytoplasm. Several pseudorabies virus studies have shown that mutations in this protein impair viral replication and affect neuroinvasion [55,56]. Thus, it is likely that HSV-1 gene expression can be directly interrupted by miR124, leading to the establishment of latency. More studies are underway to investigate the regulatory effect of miR124 on the establishment of latency.

## 4. Materials and Methods

### 4.1. Cell Proliferation

The human DRG neuronal cell line HD10.6 cell line was essentially described [18,19,20,21]. HD10.6 is cultured on a coating layer consists of 16.6 μg/mL fibronectin, and in advanced Dulbecco’s modified Eagle medium-nutrient mixture F-12 (DMEM/F12, Gibco, Waltham, MA, Cat#12634-010), supplemented with 1× B27 (Gibco™, Cat#: 17504044), 1× Glutamax, (Gibco™, #35050061), 10 ng/mL Prostaglandin E1 (Sigma-Aldrich, St. Louis, MO, Cat#: P5515), 0.5 ng/mL basic fibroblast growth factor (β-FGF), and 50 μg/mL G418 solution (Roche, Basel, Switzerland, Cat#: 04727878001).

### 4.2. Cell Differentiation

The protocol was published [19] with minor modifications. After the cells reached confluency, approximately 80%, the cells were transferred to a plate pre-coated with 2% Matrigel (Corning, Corning, NY, Cat#: 356234) and 10 μg/mL Poly-D-lysine (Advanced BioMatrix, Carlsbad, CA, Cat#5049), in maturation complete media, which comrpised Neurobasal™ Plus Medium (Gibco™, Waltham, MA, # A3582901), with 1× B-27™ Plus (Gibco™, Waltham, MA, #A3582801), 1× Glutamax, (Gibco™, #35050061), supplemented with 50 ng/mL of β-NGF (Peprotech, Rocky Hill, NJ, Cat#450-01), 25 ng/mL of GDNF (Gibco™, #PHC7041), CNTF, and NT-3 (Peprotch, #450-13, 450-03, respectively), 1 μg/mL of Tetracycline (Sigma-Aldrich, St. Louis, MO, Cat#: T-7660) and 25 μM of Forskolin (Cayman Chemical, Ann Arbor, MI, Cat#66575-29-9), in a density of 15,544 cells/cm2 for 7 days media change for every 3–4 days.

### 4.3. Introduction of miR-124 into HD10.6 Cells

A PCR-amplified 496-bp fragment from the rat genomic DNA (NC_051350) containing miR-124 (accession no. MIMAT0004728) was cloned into a lentiviral vector (pCDH-EF1-MCS-IRES-GFP). The resulting vector is called L124, tagged with GFP. miR-124 was introduced to the proliferating HD10.6 cells in a setting of 12-well plates by L124 transduction at MOI of 4, in which approximately 98% of the cells were infected [22]; the cells are called HD-L-Stable. The HD-L-Stable cells were subjected to differentiation for 5–7 days followed by HSV-1 infection.

### 4.4. Infection of HD10.6 by HSV-1

Differentiated HD10.6 cells were infected with the viral construct KVP-mRFP HSV-1 [57], consisting of the Kos strain with RFP under VP26, which was gifted by Dr. Prashant Desai, Johns Hopkins University. The cells were infected at an MOI of 0.15 or 1 for 1 h and then replaced with the infection mix by fresh media. The protocol was described previously [18], and the only exceptions were the virus strains and incubation without ACV.

### 4.5. Electrophysiology

Fluorescent HD10.6 cells were observed with a Nikon Eclipse Ti inverted microscope (Nashua, NH) equipped with Hoffman optics and epifluorescence filters to assess the infected HD10.6 cells by GFP expression [18]. Whole cell recording of voltage-activated sodium currents was performed as previously described [18]. Cell capacitance was analyzed and normalized as previously described [58,59].

### 4.6. RNA Extraction and Reverse Transcription

Total RNA extraction was performed using miRNeasy Kits (Qiagen, Germantown MD, Cat#217004) and followed by the protocol. The reverse transcription of mRNA is performed using iScript™ Reverse Transcription Supermix (Bio-Rad, #1708841) and by following the instructions of the protocol. The reverse transcription of microRNA (miR) is performed using miRCURY LNA SYBR^®^ Green PCR Kits (Qiagen, Germantown MD, Cat#339345). It is noted that we also used a discontinued product miScript^®^ II RT Kit, # 218193) in some of our results.

### 4.7. Quantitative PCR (qPCR)

The calculation of the viral gene expression by qPCR was previously described [19]. qPCR was performed using SsoAdvanced™ Universal SYBR^®^ Green Supermix (Bio-Rad, Hercules, CA, Cat# 1725271) and by following the instructions of the protocol. The quantification of gene expression was determined by arbitrary units normalized to peptidylprolyl isomerase A (PPIA) expression [18]. The primer sequences of PPIA, TK, ICP0, LAT, miR124, and miR26a are listed in Table 1.

### 4.8. Plaque Assay

The plaque assays were performed in confluent 96-well plates grown with a Vero cell line [22]. Briefly, the lysates and the supernatants were collected from infected HD10.6 cultures on triplicate sets at different days post-infection (dpi). The lysates were subjected to three freeze–thaw cycles and then spun down, forcing the virions to escape from the cell debris. The cell lysates and the supernatants from infected cultures were serially diluted before being added to Vero cell monolayers in confluent 96-well plates. The cells of the 96-well plate were infected with 50 μL/well of supernatants or lysates for 1 h, followed by several washes and the addition of fresh media. The plates were subjected to a high-throughput imaging analysis [60].

### 4.9. Data Analysis

Statistical analyses were performed with Student’s unpaired *t*-test or one-way ANOVA followed by post hoc analysis using Tukey’s honest significant difference test (STATISTICA software, Tulsa, OK, USA). All data values are presented as mean ± SEM, and *p* ≤ 0.05 was regarded as significant.

## 5. Conclusions

In conclusion, a quiescent state of infection, similar to latency, was accomplished by introducing the human miR-124 into the human DRG-derived neuronal HD10.6 cells, followed by HSV-1 infection in the absence of ACV. The mechanism of miR-124 induction of a latent-like state remains to be fully investigated. It is hypothesized that miR-124 fosters neuronal identity. It is not unlikely that HD10.6 cells, during the differentiation, failed to express certain neuronal genes, thus weakening the inhibitory effects of neuronal suppression on HSV-1 replication. The overexpression of miR-124 compensated for the deficit of neuronal expression, promoting HSV-1 latency.

## Figures and Tables

**Figure 1 pathogens-11-00803-f001:**
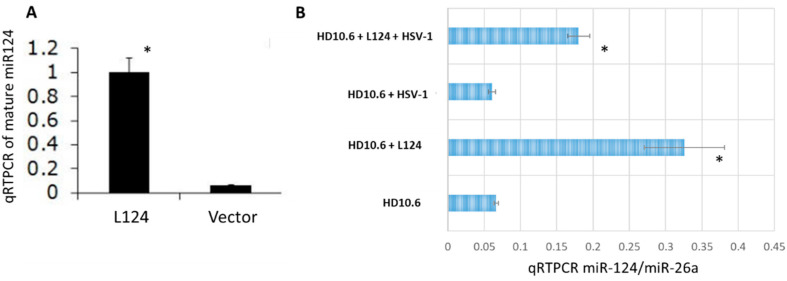
miR-124 is overexpressed in HEK-293 and HD10.6 cells following L124 transduction, performed in triplicate. The mature miR-124 was introduced into HEK-293 cells by L124 infection. The miR-124 expression level was validated by qRT-PCR. miR-124 expression appeared to increase by approximately 10-fold (**A**). HD10.6 cells were first transduced with L124 followed by HSV-1 infections at MOI of 1. The miR-124 expression was analyzed by the same method described in A. Quantitative analyses of miR-124 expression indicate a 3-fold and 4.95-fold upsurge with or without HSV-1 infection, respectively (**B**). All qRT-PCR experiments were normalized to miR-26a in a quadruple fashion for statistical analyses. The asterisk (*) designates significant differences compared to vector or no L124 control. Note that both A and B were normalized to miR-26a. The viral titer of the Figure 1B supernatant is 3.6 × 103 ^PFU/mL^ for the HD10.6 + HSV-1 group, while the others are less than 100 PFU/mL (determined by standard plaque assay; data not shown).

**Figure 2 pathogens-11-00803-f002:**
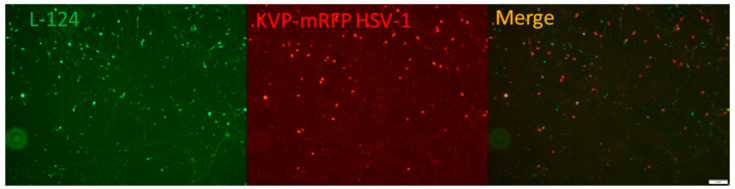
Differentiated HD10.6 cells with miR124 overexpression can be co-infected by HSV-1. The L124-transduced HD10.6 cells (HD-L-Stable) were subjected to KVP-mRFP HSV-1 infection at the MOI of 1 followed by fluorescent microscopy. The green and red fluorescence represented the miR124 expression and HSV-1 infection, respectively. The merged image demonstrated the presence of miR-124 in the HSV-1 infected cells. The size of the scale bar is 1 mm.

**Figure 3 pathogens-11-00803-f003:**
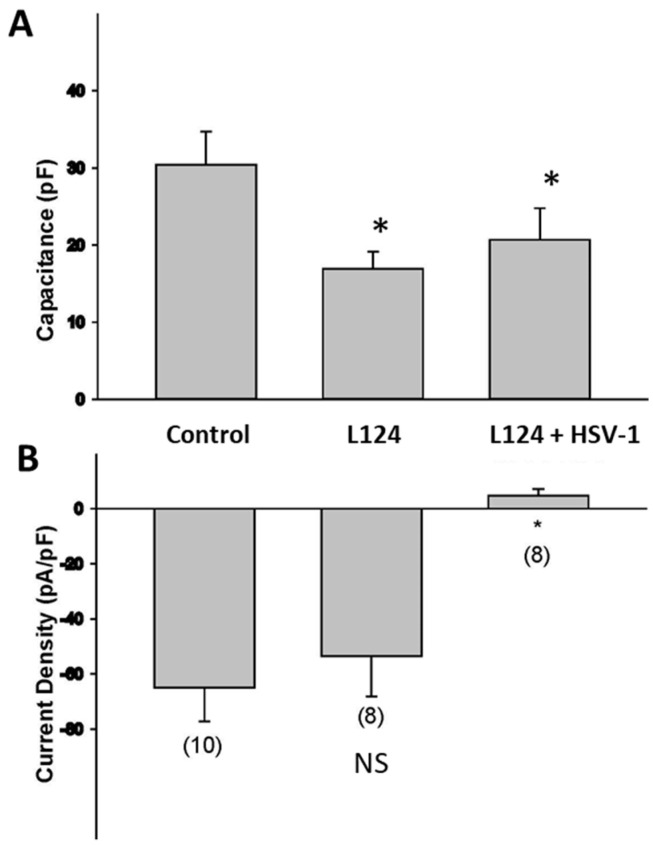
Electrophysiology of infected HD10.6 cells following miR-124 overexpression. Electrophysiological studies indicated that overexpression of miR-124 evokes a 47% decrease in cell capacitance, suggesting a decrease in the size of HD10.6 cells. The *y*-axis denotes Capacitance (pF) (**A**). Analysis of the sodium current densities demonstrated no significant difference between control and HD-L-Stable cells. HSV-1 acute infection decreases the functional expression of sodium channels. The *y*-axis denotes current density (pA/pF) (**B**). NS: denotes “Not significant”; * denotes statistically significant differences.

**Figure 4 pathogens-11-00803-f004:**
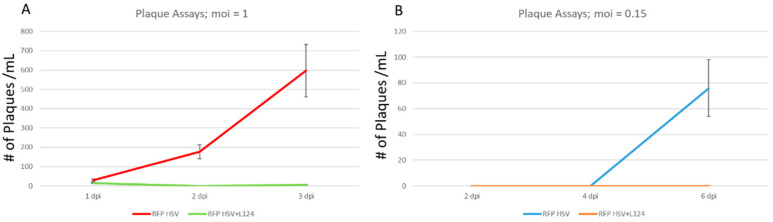
A dormant state of HSV-1 infection was achieved with miR-124 overexpression. Differentiated HD10.6 cells, with or without L124 transduction, were subjected to HSV-1 infection, in the absence of ACV, at the MOI of 1 followed by plaque assays using the media supernatant collected daily for three days. It is visible that HSV-1 replicated actively in HD10.6 cells without miR-124. The HD-L-Stable cells, however, released fewer viral particles (**A**). Other infections were performed with the MOI of 0.15 followed by the same analyses. It appeared that the infectious viruses were not released until 6 dpi in regular HD10.6 cells, and no release of infectious virus was detected in HD-L-Stable cells. Note that the assays were measured at 2, 4, and 6 dpi due to much lower titers used for infection (**B**). All plaque assay experiments were performed in triplicate for statistical analyses.

**Figure 5 pathogens-11-00803-f005:**
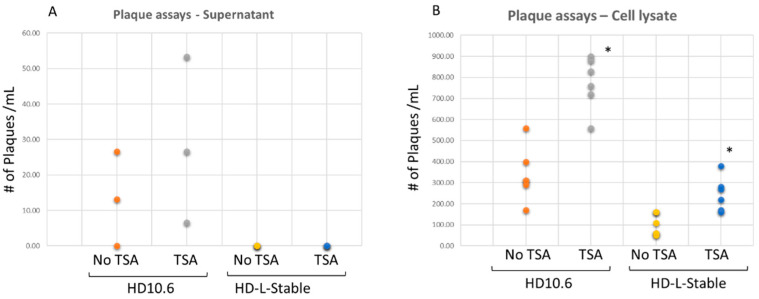
HSV-1 was not released but present within miR124 overexpressing HD10.6 cells. HD-L-Stable cells in a latency-like state were subjected to reactivation by the histone deacetylase inhibitor TSA. No release of infectious viruses was detected in the media supernatant of the HD-L-Stable cells, even after TSA treatment. In the regular HD10.6 cells, infectious viruses from the media supernatant were spotted in two out of three wells and the number increased after TSA treatment (**A**). The cell lysate was subjected to plaque assays, and the infectious viral particles were observed in both cases, but HD-L-Stable cells exhibited approximately a 3.8-fold reduction in plaque formation compared to the original HD10.6 cells (compare orange to yellow spots). The TSA was sufficient to boost the viral replication in both cases. All plaque assay experiments were performed in sextuplicate followed by dot plot analyses and the statistical analyses by ANOVA indicated the differences were significant (denoted by *) (**B**).

**Figure 6 pathogens-11-00803-f006:**
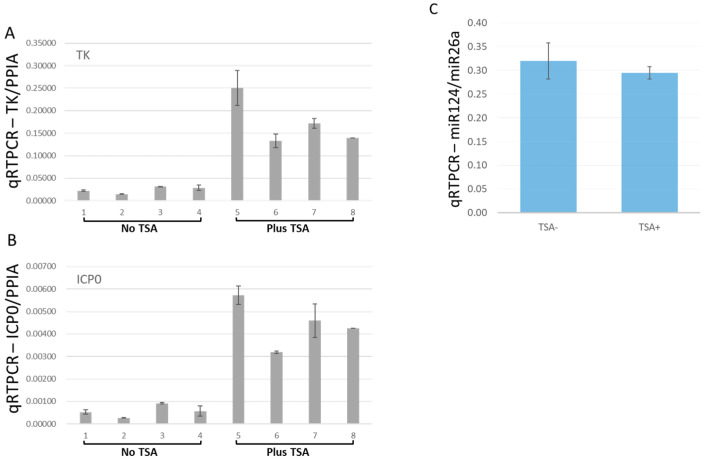
Viral transcription increased after TSA-induced reactivation from the dormant infection. The effects of TSA to reactivate the viral gene expression from HD-L-Stable cells were measured by qRT-PCR. The transcripts of TK were quite weak but increased approximately 7.2-fold after TSA treatment (**A**). The ICP0 transcription exhibited a similar pattern with 7.8-fold increases upon TSA reactivation (**B**). The miR-124 expression maintained a similar level after the treatment (**C**). Note that, in (**A**,**B**), the gene induction by TSA varied probably due to the different copy numbers of viruses maintained during latency.

**Figure 7 pathogens-11-00803-f007:**
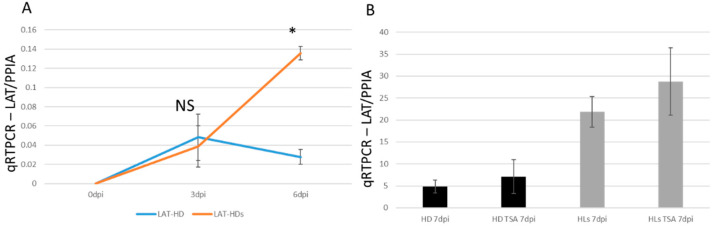
Viral transcription increased after TSA-induced reactivation from the dormant infection. The HSV-1 induced gene LAT accumulates in HD10.6 cells overexpressing miR-124. The accumulation of HSV-1 LAT was analyzed by qRT-PCR. No difference was observed at 3 dpi but a significant increase appeared in HD-L-Stable cells at 6 dpi (orange line). The LAT accumulation in regular HD10.6 cells is displayed as a blue line (**A**). The accumulation of LAT showed no difference after TSA treatment (**B**). * denotes statistically significant differences.

**Figure 8 pathogens-11-00803-f008:**
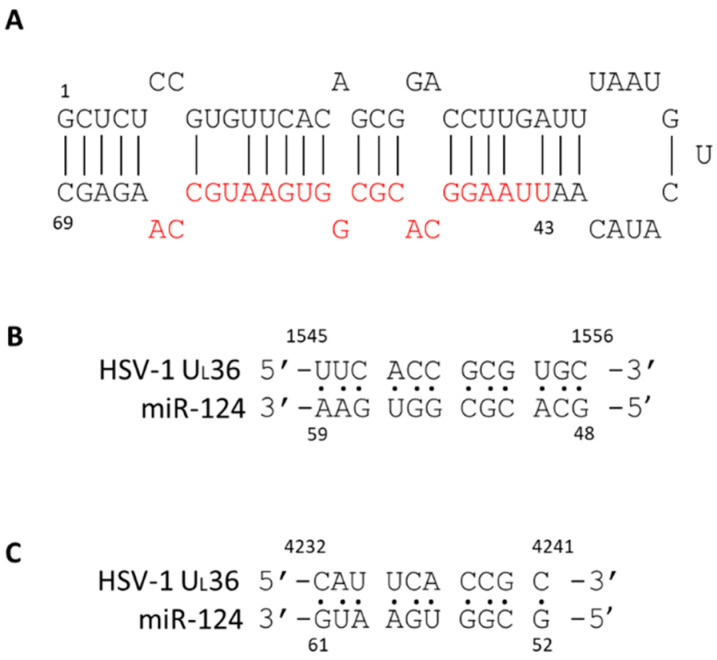
Structure of miR-124 and the putative binding to HSV-1 U_L_36. The stem-loop structure of miR-124 was depicted. The functional domain was labeled in red (**A**). The putative match by computational analyses was described and the results exhibited two hits with 100% match. The miR-124 was first matched from 48–59 to U_L_36 cDNA 1545–1556 (**B**). The second match of miR-124 was from 52–61 to U_L_36 cDNA 4232–4241 (**C**). The HSV-1 sequence is based on the strain McKrae (GenBank#: MN136524.1).

**Table 1 pathogens-11-00803-t001:** PCR primer sequences and annealing temperatures.

Target [gene]	Primer Sequence	Annealing Temp
miR-26a	CGAGTTCAAGTAATCCAGGA (f); CCAGTTTTTTTTTTTTTTTAGCCTATC (r)	60 °C
miR-124	TAAGGCACGCGGTGA (f); CCAGTTTTTTTTTTTTTTTGGCAT (r)	60 °C
PPIA(h]	AGCATACGGGTCCTGGCATCT(f); CATGCTTGCCATCCAACCACTCA(r)	65 °C
GFP	ACTTCAAGATCCGCCACAACA (f); TGATCGCGCTTCTCGTTGG (r)	58 °C
ICP0 (v)	GACGGGCAATCAGCGGTTC(f); GTAGTCTGCGTCGTCCAGGT(r)	60 °C
TK (v)	TACCCGAGCCGATGACTTAC(f); AAGGCATGCCCATTGTTATC(r)	62 °C
LAT (v)	CGGCGACATCCTCCCCCTAAGC(f); GACAGACGAACGAAACGTTCCG(r)	60 °C

## Data Availability

Not applicable.

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
