# Peer review of "Establishing a Herpesvirus Quiescent Infection in Differentiated Human Dorsal Root Ganglion Neuronal Cell Line Mediated by Micro-RNA Overexpression"

_pathogens, 2022, doi:10.3390/pathogens11070803_

Round 1

Reviewer 1 Report

Chen and colleagues have developed a human DRG neuronal cell culture model HD10.6 to study HSV-1 latency and reactivation. They then assessed the role of the micro RNA 124 in the establishment of the HSV-1 latency.  

Major comments :

- The authors should add to the results section several details about the reagents and techniques used in order to facilitate their understanding:

-        -  In chapter 2.1, it should be specified what L124 corresponds to. Also, the term infection should be replaced by transduction since it is a lentivirus. The authors do not indicate in the materials and methods how the transduced cells were then selected. They also do not indicate that the transduced HD10.6 cells become the HD-L-Stable cells. Finally, the authors do not explain why they studied the expression of miR124 in cells infected with HSV-1, whether transduced or not.

-       -   In chapter 2.2, authors should indicate what the KVP-mRFP HSV stands for.

-        -  In chapter 2.3, the title indicates a reduction in the size of HD10.6 cells by miR124 but the conclusion of the chapter is that infection by HSV-1 reduces the density of sodium channels. What is the link with the title of the chapter and what is the consequence of this findings? Does miR124-induced size reduction indicate cell differentiation? If not, what are the markers used to demonstrate that miR124 induces neuron differentiation?

-      -  In chapter 2.7, authors should specify the MOI used during the LAT accumulation experiments since the MOI seems to influence the course of the infection, productive or not.

The title and conclusions of certain chapters of the results are not demonstrated by the experiments carried out:

-         -  In chapter 2.4, the results presented suggest an inhibition of replication by overexpression of miR124 without evidence of initiation of latent infection. The absence of latency in HD-L-Stable cells is demonstrated in Figure 5 where infectious viruses are detected in the cells which is not the definition of viral latency. The authors could also discuss why the dormant infection is not reached at the MOI of 1.

-        - The use of TSA to induce viral reactivation is questionable. It is not certain that the virus is actually latent in the treated cells. There is also no evidence that the action of TSA to induce reactivation specifically antagonizes the mechanism induced by miR124 and/or will be more potent to induce ractivation than that of miR124 to induce latency. Also, how to explain that virus progeny are trapped within the cells after TSA treatment ? This suggests a role of blocking exocytosis by miR124 rather than a role of inducing latency to explain the absence of virus detection in the culture supernatant. Finally, the authors do not indicate whether the differences in viral quantification in HD-L-stable cells treated or not with TSA were significant.

Minor comments :

-        - The authors indicate in discussion line 180 that only non-coding viral RNAs are detectable during HSV1 latency. To my knowledge, several genes, other than LATs, are expressed during the latency phase: ICP27, UL23…

- The paper was not properly edited for typos (see legends of the figure 7).

Author Response

We thank the Reviewer 1 for the critiques and they are quite helpful. Please see the attachment for point-to-point responses.

Reviewer 2 Report

The paper by Chen et al. described human neuronal cell line model where the overexpression of miR124 in HD10.6 neuronal cells induces latency of HSV-1 in these cells. The authors suggest that miR124 may play a role in the establishment of latency of HSV-1 in DRG neurons.

Please see below for minor comments that need to be addressed:

Figure 1 – these two graphs both present PCR data but are presented in different ways. Can Fig 1B be changed to match the rest of the PCR graphs in the manuscript for consistency?

Figure 2 – increase size of scale bar - the current scale bar is very difficult to read

Lines 86 - 94: please clarify. How does a decrease in the cell capacitance following miR124 overexpression correlate to the cell size?

Figure 3 – increase text size, particularly size of y-axis

Figure 6 – what do the numbers on the x-axis represent? Are these biological replicates? And if so, why aren’t they averaged and presented as a single bar?

Figure 7A – blue and orange bars labels are not defined – keep labels consistent with previous graphs. Significance is not specified

Please include sample sizes for Figs 1 and 3

Figs 1, 6, 7 – x-axis title is not appropriate – what are you graphing here? Are these absolute or relative copy numbers?

Lines 291 - 294 PCR methodology – is PPIA used as a reference gene? This needs to be stated somewhere in the methods section. Mention of controls is also lacking. Were no template controls and -RT controls included in the PCR runs?

Figure 2 – methodology for this data is missing. Is L-124 tagged to GFP? Or was this staining from antibodies? Methodology on image acquisition is needed.

Author Response

We appreciate the comments from Reviewer 2. These are critical issues we need to address. Please see our point-to-point responses in attachment.

Reviewer 3 Report

Chen and colleagues try to demonstrate a correlation between miR-124 and latent/productive HSV-1 infection in an in vitro HSV-1 infection model involving human DRG neuronal cells (HD10.6) differentiated and transfected with a lentivirus overexpressing miR-124.  Their study is promising but a specific demonstration of what is claimed is lacking. In addition, the text is not easy to read, because data are often missing and, also, the several paragraphs interrupt the attention of the reader who realizes that something is missing and this something is surprisingly found in the next paragraph. I am sorry but  the paper cannot be considered for publication. However, some considerations are listed below.

From Line 60: results immediately start with the appearance of L124 without saying what it is. I had to look through the methods to understand. Please edit. The same is observed in all the results, because probably the paper was originally structured with the methods first and the results later. Unfortunately, this makes everything difficult to read, so the results have to be changed

Fig 1 B: In HSV-1 infected sample, miR-124 levels are identical to those in the uninfected sample. This means that acute HSV-1 infection of these cells does not induce an increase in miR-124, nor its reduction. Why does the virus reduce miR-124 levels during its acute infection in the sample also infected with L124? What is the viral title in these samples in PFU/ml (standard plaque assay)?

 In addition, in A the y-axis shows qRTPCR of mature miR-124, while in B it shows qRTPCR of miR-124 normalized to miR-26a; in the caption of the figure the authors say the analysis performed is the same. Was it also normalized in A as in B?

Fig 2: The authors show HD-L-Stable cells infected with KVP-mRFP HSV, but do not mention viral titer.  A comparison analysis with HD10.6 cells (differentiated or not) is missing: is the titer the same? Is HSV-1 replication affected by miR-124? No such data come out from this figure. No data about time of infection is reported. In addition, the authors cite reference No. 20 for HD10.6 cells cell differentiation and infection protocol; in reference No. 20, infections were performed with the HSV-1 strain McKrae while here HSV-1 strain Kos was used; kos and McKrae have different virulence.  Are the authors sure that their strain is equivalent to the  used in ref. 20?

Fig.4 and 5: The intracellular virus analysis is not convincing; in particular, what is not convincing is the use of TSA that is taken for granted. Have the authors considered previous work? One gets the impression that all this work was inspired by reference no. 20 (https://doi.org/10.3389/fmicb.2016.01970; doi: 10.1080/13550280590952817)

Author Response

Although not enthusiastic about this manuscript, we still appreciate the critiques from the Reviewer 3. We have carefully addressed each comment in attachment and hope it will clarify the concerns.

Round 2

Reviewer 3 Report

I appreciate the authors' responses but unfortunately disagree on some points, as listed below:

First round: Fig 1 B: In HSV-1 infected sample, miR-124 levels are identical to those in the uninfected sample. This means that acute HSV-1 infection of these cells does not induce an increase in miR-124, nor its reduction. Why does the virus reduce miR-124 levels during its acute infection in the sample also infected with L124? What is the viral title in these samples in PFU/ml (standard plaque assay)?

Response: This is a very interesting question and we don’t have a solid answer. The miR-124 we overexpressed is amplified from rats. The sequence of functional domain is identical between human and rats but the rest of the sequence may be slightly different. We realized that HSV-1 has a tegument protein called virion host shutoff (VHS) which is an RNAse that triggers the degradation of mRNA as well as dsRNA. It is likely that the L124-expressed miR-124 is less resistant to VHS comparing to the endogenous one, thus showing the observation. By the way, the virus titer used in this experiment is at moi of 1.

Second round: I may not have been able to explain my question well, and I apologize for that. Let me try to rephrase it: the infections at the beginning were done at 1 moi, but what I asked is different. I asked what the viral titer in PFU/ml of each sample was at the end of the experiment. This information is indispensable and cannot be omitted.

Second round : Line 65: miR-124 overex- 65 pression did not affect the HSV-1 infection (data not shown).

I need to know how viral titer was calculated (i.e.  standard plaque assay or TCD50 or other specific test) and the results in pfu/ml to understand the titer of infection.  Please add these data in the manuscript.

Second round:  Figure 5B: “HD-L-Stable cells exhibited approximately a 3.8-fold reduction in plaque formation compared 152 to the original HD10.6 cells (Compare orange to yellow spots)”

I don’t see this fold reduction. In the graph, the viral titer of “orange samples” is included between 5,5x10e5 and 1,8x10e5 PFU/ml; the viral titer of “yellow samples” is under 1,8x10e5  PFU/ml. The authors didn’t consider that the viral titers are similar, contained in the same log! A viral reduction is effective when, for example, a sample has 5,5x10e5 PFU/ml and the other sample has 5,5x10e3, because viral reduction is logarithmic. Please edit your results according to this basic but mandatory virology rule.

Author Response

We appreciate the reviewer's opinion and felt it is quite helpful. The point-by-point response to the reviewer’s comments is attached.

Round 3

Reviewer 3 Report

I thank the authors for the changes and modifications made.